# The Inhibitory Effects of RNA-Interference-Mediated Guanylate Cyclase Knockdown on Larval Metamorphosis and Early Progeny Growth of Razor Clam

**DOI:** 10.3390/genes14020459

**Published:** 2023-02-10

**Authors:** Yuting Han, Beibei Li, Yifeng Li, Donghong Niu

**Affiliations:** 1Shanghai Collaborative Innovation Center for Cultivating Elite Breeds and Green-Culture of Aquaculture Animals, Shanghai Ocean University, Shanghai 201306, China; 2Key Laboratory of Exploration and Utilization of Aquatic Genetic Resources, Ministry of Education, Shanghai Ocean University, Shanghai 201306, China

**Keywords:** *Sinonovacula constricta*, guanosine cyclase, metamorphosis

## Abstract

Guanylate cyclase (*GC*, *cGMPase*) is a key enzyme in organisms, catalyzing the synthesis of *cGMP* from *GTP*, thus making *cGMP* work. *cGMP* plays a vital role in the regulation of cell and biological growth as a second messenger in signaling pathways. In this study, we screened and identified *cGMPase* from the razor clam *Sinonovacula constricta*, which encoded 1257 amino acids and was widely expressed in different tissues, especially the gill and liver. We also screened one double-stranded RNA (dsRNA), *cGMPase*, which was used to knockdown *cGMPase* at three larval metamorphosis development stages: trochophores-veliger larve, veliger larve-umbo larve, and umbo larve-creeping larvae. We showed that interference at these stages significantly inhibited larval metamorphosis and survival rates. *cGMPase* knockdown resulted in an average metamorphosis rate of 60% and an average mortality rate of 50% when compared with control clams. After 50 days, shell length and body weight were inhibited to 53% and 66%, respectively. Thus, *cGMPase* appeared to regulate metamorphosis development and growth in *S. constricta.* By examining the role of the key gene in the metamorphosis development of *S. constricta* larvae and the growth and development period, we can provide some data reference for studying the growth and development mechanism of shellfish, and the results provided basic information for the breeding of *S. constricta.*

## 1. Introduction

RNA interference (RNAi) is a biological technology by which double-stranded RNA (dsRNA) induces sequence-specific gene silencing by targeting mRNA for degradation [1]. Recent studies reported that RNAi technology was widely used in several model organisms, including the *Drosophila* [2], *Apis mellifera* [3], *Blattella germanica* [4], and *Sinonovacula constricta* [5]. In recent years, RNAi was used to study pathways or gene functions in bivalves, e.g., the in vivo injection of oyvl-dsRNA into the oyster gonad caused a premature stopping of germ cell proliferation and meiosis throughout the organ, while inhibiting germ cell development [6].

GC/cGMP is an important signal transduction pathway in organisms is closely related to the development of organisms. In this transduction pathway, guanosine cyclase (*GC, cGMPase*) is a key porphyrinase in vivo. In 1969, *cGMPase* was found to have catalytic capacity. In the body, *cGMPase* can catalyze guanosine triphosphate (GTP) and promote the synthesis of cyclic guanosine monophosphate (cGMP) from GTP [7,8]. Moreover, the activity of *cGMPase* can directly affect the level of cGMP, and the concentration of cGMP directly affects other diseases [9]. There are few studies on the GC/cGMP signaling pathway in bivalves. In marine bivalves, G-protein-coupled receptors (GPCRs) and signal transduction by the GC/cGMP pathway could mediate the metamorphosis of larvae in the mussel *Mytilus edulis* [10].

cGMP belongs to a class of cyclic nucleotides first discovered in rat urine [11]. The molecule is a second messenger that transmits intracellular messages, activated by protein kinases activated by G-protein-coupled receptors, to transmit extracellular signals to the nucleus. cGMP signaling involves a series of signal transduction pathways, such as the cGMP-PKG signaling pathway [12], mitochondrial NO-cGMP pathway [13], and light sensing effects in vertebrates [14]. cGMP is also involved in glycogen decomposition [15], opening ion passages into and out of cell membranes [16], apoptosis [17], and relaxing smooth muscle [18].

In China, *S. constricta* is an important economically cultured shellfish. As a bivalve, larval rearing is the most important period and is an important factor determining final breeding yields. Metamorphosis development is a vital physiological process; therefore, it is particularly important to study the molecular mechanisms underpinning *S. constricta* larval development. Previously, high-throughput sequencing technology was used to identify key transcripts in *S. constricta*; in particular, *cGMPase* was identified as having a key involvement in cell growth and development during morphological development stages, including trochophore, veliger, and creeping larvae stages [19].

Recent studies reported that GC/cGMP signaling mediated the attachment and deformation of invertebrate larvae, such as *Balanus Amphitrite* [20], the sea slug *Alderia Willowi* [21], and *Bugula Neritina* [22]. To explore if *cGMPase* was involved in *S. constricta* larvae regulation and growth processes, we use RNA interference to knock down *cGMPase* in *S. constricta* developmental stages. The effects of *cGMPase* gene on growth and development were investigated by analyzing the metamorphosis rate and survival rate of *S. constricta*. These results will provide a theoretical basis for understanding the molecular mechanisms underlying *S. constricta* larvae development and provide valuable signal transduction gene information for growth processes in marine shellfish.

## 2. Materials and Methods

### 2.1. Study Animals and Sample Preparation

The study followed the protocol for the treatment of experimental animals from the Institutional Animal Care and Use Committee (IACUC) of Shanghai Ocean University, Shanghai, China. Razor clams are produced in the Donghang Breeding Base in Zhejiang Province, China. Healthy adult clams were selected and placed into net cages (volume 1 m × 0.5 m × 0.2 m) suspended in breeding ponds. To create a dark environment, shade nets were used to cover cages to stimulate spawning.

### 2.2. Total RNA Extraction and cDNA Synthesis

Total RNA was extracted using an animal tissue total RNA extraction kit (TIANGEN, Beijing, China) following manufacturer’s instructions. RNA concentration and purity were determined using the Nanodrop 2000C spectrophotometer (Thermo Fisher Scientific, Waltham, MA, USA). RNA integrity was confirmed using 1% agarose gel electrophoresis. Extracted total RNA was transcribed using the PrimerScript RT kit (Takara, Japan) according to manufacturer’s instructions. cDNA was then stored at −20°C.

### 2.3. Verification of cGMPase Sequences

*cGMPase* fragment sequences were obtained from the razor clam transcriptome library [19]. Specific *cGMPase* fragment primers were designed using Primer 5 software. Amplified products were verified by 1% agarose gel electrophoresis, recovered, and purified using the MiniBEST Agarose gel DNA extraction kit Ver. 4.0 (Takara) according to manufacturer’s instructions. Purified products were ligated into the PMD19-T Vector at 16 °C for 1 h, transformed into DH5-α competent cells, selectively plated onto agar plus ampicillin, and incubated overnight at 37 °C. Positive colonies were cultured in LB medium plus ampicillin for 4 h; positive clones were confirmed by PCR using universal primers (Table 1) and the bacterial solution as template. The bacterial solution with the expected band size was selected and sent to Sangon Biotech (Shanghai, China) for sequencing. After removing the vector sequence, the sequencing results were compared with the original sequence.

### 2.4. Sequencing and Phylogenetic Analyses

Amino acid sequences were retrieved using open reading frame (ORF) analysis (ORF Finder (https://www.ncbi.nlm.nih.gov/orffinder/, accessed on 10 February 2022)). Predicted amino acid sequences were confirmed and analyzed for sequence homology using the BLAST algorithm (http://www.ncbi.nlm.nih.gov/blast, accessed on 10 February 2022). Protein physicochemical parameters were predicted by ProtParam (https://web.expasy.org/protparam/, accessed on 10 February 2022). SignalP was used to identify the protein signal peptide (http://www.cbs.dtu.dk/services/SignalP-2.0/, accessed on 10 February 2022). The TMHMM server V2.0 predicted transmembrane structures (http://www.cbs.dtu.dk/services/TMHMM/, accessed on 10 February 2022). Phosphorylation sites were analyzed in Netphos 3.1 (http://www.cbs.dtu.dk/services/NetPhos/, accessed on 10 February 2022). Protein domain positions and types were predicted by SMART (http://smart.embl-heidelberg.de/smart, accessed on 10 February 2022).

### 2.5. qRT-PCR of cGMPase mRNA

Total RNA was extracted from the liver, gill, foot, hemolymph, mantle, gonad, and siphon of adult razor clams and from different larval stages (embryos, trochophores, veliger larvae, umbo larvae, creeping larvae, and juveniles) and stored at −80 °C. According to manufacturer’s instructions, RNA was reverse-transcribed using the PrimeScriptTM RT reagent kit (Takara, Japan). Primer 5.0 software was used to design specific primers for *cGMPase*. As 18S rRNA is equally expressed in adult tissues [23,24], embryos or other stages, we used 18S rRNA as a housekeeping gene (Table 1). TB Green Premix Ex Taq II (Takara, Dalian, China) was used for relative gene expression analysis in a CFX96 system (Bio-Rad, Hercules, CA, USA). Final qRT-PCR reactions (20 μL = final volume) comprised 1.6 μL cDNA, 10 μL 2 × SYBR Premix Ex Taq™ (Takara, Japan), 0.8 μL each primer (10 μmol/L), and 6.8 μL ddH_2_O. The following parameters were used: one cycle of 95 °C for 30 s, followed by 35 cycles of 95 °C for 5 s, 55 °C for 30 s, and 72 °C for 1 min, and a final cycle of 72 °C for 3 min. Melting curve analyses were performed and the data were analyzed using the 2^−ΔΔCt^ method.

### 2.6. Synthesizing and Screening Effective dsRNA against cGMPase

#### 2.6.1. Processing PCR Products

Three dsRNA interfering chains (dsRNAs) were designed within three pairs of PCR primers. Then, T7 promoter sequences were added to upstream and downstream 5′ ends of the three pairs of PCR primers. T7 + F1, R1 and F1, T7 + R1 were used as upstream and downstream primers and were amplified by ordinary reverse transcription PCR twice. These primers are shown (Table 2).

Following the manufacturer’s instructions (TransGen Biotech, Beijing, China), 1 μg cDNA template, 4 μL 5 × T7 transcription buffer, 8 μL 10 m MNTP mix, and 2 μL T7 transcription e-mix were added to RNAse-free water and made up to 20 μL. After mixing, transcription was set at 37 °C for 2 h.

DNA was then removed and annealed to form dsRNA, while single-stranded RNA (ssRNA) was removed. Both PCR products were mixed at the same concentration on ice and reacted at 70 °C for 10 min. Samples were removed immediately after transcription and left at room temperature for 20 min. Diluted RNase solution (1 μL) was added to each 20 μL volume and incubated at 37 °C for 30 min. Finally, samples were purified as described.

#### 2.6.2. dsRNA Purification

To samples, an equivalent volume of isopropyl alcohol and 1/10 the volume of 3 M sodium acetate was added, mixed, and incubated on ice for 5 min. Samples were then centrifuged at 12,000 rpm at 4 °C for 10 min, mixed until a white precipitate appeared, and 500 μL 70% ethanol diluted in enzyme-free water was added. Samples were washed twice at 12,000 rpm at 4 °C for 10 min and air-dried at room temperature for 15 min, after which 2–5-times enzyme-free water was added to dissolve. Concentrations were determined and samples were stored at −80 °C.

### 2.7. Interference Studies during Larval Development

Prior to interference studies, preliminary experiments were designed to explore effective dsRNAs synthesis against *cGMPase*. First, immersion experiments were performed to determine suitable effective dsRNAs. Second, concentrations of effective dsRNAs were screened, and five concentration gradients were established: 5 ng/μL, 10 ng/μL, 25 ng/μL, 50 ng/μL, and 100 ng/μL.

*S. constricta* larvae were selected from veliger larve (DVE) to umbo larve (UMBO) metamorphosis processes, and included five dsRNA concentration groups, a negative control (NC) group (inject DEPC water to rule out the effect of injection on razor clam), and a blank control group (without any treatment). Larvae were cultured in six-well plates and three replicates were performed for each group. (Each hole height of the culture plate is about 1.75 cm, and the effective culture area of each hole is about 9.6 cm^2^, so the volume is 16.8 mL. Each well contained about 100 larvae and 10 mL of total culture water.) Plates were observed until larval metamorphosis, and both blank and negative control groups were particularly observed. When complete metamorphosis (>80% metamorphosis rate) was achieved in the negative control group, larval metamorphosis and survival rates were examined by microscopy and recorded. During the calculation of mortality, a fixed amount of larvae mixture was absorbed from each well, counted under a microscope, and then the number of dead individuals/the total number of individuals with mixed liquid was calculated. After observation, the larvae were put back into the corresponding culture well and absorbed again. The operation was repeated three times. After the optimal dsRNA concentration was identified, the same immersion experiments were performed at the other two larval stages: trochophores (TPH)-DVE and UMBO-creeping larvae (CRE) to test dsRNA interference on *S. constricta* larvae metamorphosis and survival rates.

### 2.8. Adult Growth Test of dsRNA

dsRNA was injected into *S. constricta* with shell length of about 3 cm. Control, negative, and experimental groups were established. Thirty clams, with the same specifications, were selected for each group in triplicate. Repetitions were performed in a water tank (85 cm × 55 cm × 40 cm). For these studies, 100 μL of *cGMPase* gene with 500 ng/μL interference was injected into each clam. Due to the long duration of the growth experiment, time points were established, and samples were removed at 6, 12, 24, 48 h, and 3, 4, 5, and 7 days after injection. It provides data support for dsRNA supplementary injection and continuous interference in subsequent formal growth experiments. Sampling was used for subsequent quantitative studies.

Razor clam growth indices (shell length and body weight) were measured and recorded using a Vernier caliper and balance, respectively, on days 0, 25, and 50 during the study.

### 2.9. Statistical Analysis

The experimental data were presented as the mean ± standard error. One-way ANOVA was performed using SPSS 22.0 (Chicago, IL, USA). Differences between the different treatment groups were tested using *t*-test for statistical significance (*p* < 0.05) and graphs were created using Sigmaplot 12.3.

## 3. Results

### 3.1. cGMPase Sequence Analysis

One *cGMPase* sequence was obtained from the razor clam. The Open Reading Frame (ORF) was 3774 bp and encoded 1257 amino acids. The predicted molecular weight was 143.11 kDa and the theoretical isoelectric point was 5.71. Razor clam *cGMPase* was a non-transmembrane protein with no peptide signal. Amino acid composition analysis showed the protein contained 21 serine phosphorylation sites (S), 9 threonine phosphorylation sites (T), and 10 tyrosine phosphorylation sites (Y). Tertiary structural predictions showed the protein comprised 40% α-helix, 15% β-strand, 4% Transmembrane helix, and 13% disordered structures.

### 3.2. cGMPase Amino Acid Sequence Homology and Phylogenetic Analysis

For *cGMPase* amino acid sequence alignment and evolutionary tree analysis, *cGMPase* amino acid sequences from twelve representative species were selected from the National Center for Biotechnology Information database. The results showed that *cGMPase* from different species had a large overlapping region (black shading), which was hypothesized to be a conserved structure (Figure 1). The *cGMPase* phylogenetic tree was composed to two main clusters, indicating that two main isoforms of the enzyme exist (Figure 2). Phylogenetic tree analysis showed that the razor clam was clustered into a separate group in the evolutionary tree; it was more closely related to *Mercenaria mercenaria* and *Pecten maximus*. Mammals and fish were more closely related. 

### 3.3. cGMPase Expression Analysis in Different Tissues

Using qRT-PCR, *cGMPase* was detected in all tested tissues, with the highest expression in the gill and liver, followed by the mantle, siphon, and foot, with lower levels in gonad and haemolymph tissue (Figure 3).

### 3.4. dsRNA Effective Sequence and Concentration

dsRNAs of *cGMPase* gene were synthesized experimentally to select and determine the appropriate effective interference concentrations. The interference levels of chain 1, 2, and 3 were 64.52%, 64.81%, and 87.28%, respectively. From a preliminary screening, chain 3 was effective against *cGMPase* (Figure 4). We observed that 50 ng/μL and 100 ng/μL concentrations optimally interfered with *cGMPase* (Figure 5). By combining statistical results from metamorphosis and mortality rates during DVE—UMBO, the larval metamorphosis rate decreased with increased inhibitory dsRNA concentrations, while the mortality rates at 50 ng/μL and 100 ng/μL concentrations were 8% and 16%, respectively (Figure 6). Thus, the effective immersion dsRNA concentration against *cGMPase* was 50 ng/μL.

### 3.5. The Effects of dsRNA on Metamorphosis

From initial experiments, at the razor clam metamorphosis stage (DVE-UMBO), the effective concentration of dsRNA toward *cGMPase* was 50 ng/μL. When compared with the control group, this concentration significantly affected the metamorphosis rate. We also investigated two other key metamorphosis development stages, TPH-DVE and UMBO-CRE. While most larvae metamorphosis occurred in the NC group, in the interference group the metamorphosis rate was significantly suppressed and caused increased mortality (Figure 7).

### 3.6. The Effects of dsRNA on S. constricta Growth

In *S. constricta*, effective interference time points were screened by the injection method. The results showed that there was little difference in the NC group at different time periods, and it was a normal fluctuation. After *cGMPase* interference, the effects were most significant at 72 h, and as time progressed, *cGMPase* expression decreased and then increased. These results suggested that other molecular compensation or related pathway regulation mechanisms may have increased gene expression after interference (Figure 8).

Additionally, dsRNA of *cGMPase* inhibited shell length and body weight growth. The increase in shell length and body weight in the NC group showed a normal trend from 0 to 50 days. When compared with controls, the dsRNA group with the gene significantly reduced the increase in body weight and shell length at day 25 (*p* < 0.05), and there were significant differences in their growth indices on day 50 (*p* < 0.05) (Figure 9).

## 4. Discussion

Metamorphosis is a common biological phenomenon in the growth and development of most invertebrates [23]. Metamorphosis is a life-stage transition characterized by high mortality rates in bivalves, so it is important to understand how metamorphosis is modulated. Attachment and metamorphosis of shellfish larvae is a key link in their life history, which lays the foundation for the breeding of larvae [24]. In previous studies, there has been less focus on the metamorphosis development of bivalves, which are all in the exploratory stage. Most of these studies are related to neurotransmitters, such as L-DOPA [25], GABA [26], serotonin, and catecholamine [27]. However, the research on the growth of *S. constricta* is still in its infancy, especially the research on the metamorphosis of razor clam larvae. Studies have shown that dopamine-hydroxylase plays a regulatory role in larval metamorphosis and growth [28].

Guanylate cyclase (*GC*, *cGMPase*) has a direct regulatory effect on cell growth, apoptosis, and vasodilation [29]. *cGMPase* is part of the G-protein signaling cascade, and its activity is negatively correlated with intracellular calcium concentration. When *cGMPase* is activated in the GC/cGMP pathway, GTP synthesis to cGMP is promoted [30]. cGMP is involved in cell signal transduction pathways, cell division, and differentiation. Therefore, the study of *cGMPase* is also a key link in the study of the role of cGMP, and the role of cGMP is directly reflected in the impact on *cGMPase*. There are few studies on the GC/cGMP pathway in bivalves, so this study aims to study the regulation of cGMP on the metamorphosis and growth of *S. constricta* by exploring gene *cGMPase*. Here, we screened the gene *cGMPase* of the razor clam and analyzed its sequence. Cluster analysis found that razor clam was a separate branch. The *cGMPase* sequence was conserved in a wide range of species, suggesting that the gene has been highly conserved during evolution.

cGMP is produced under the catalysis of *cGMPase* and guanosine triphosphate (GTP). Among them, *cGMPase* can be activated after binding to membrane receptors. Free guanylate cyclase in the cytoplasm can be activated by NO to synthesize cGMP [31,32,33]. *cGMPase* is one of the decisive enzymes for cGMP production and the only receptor for NO, so *cGMPase* is closely related to many signal transduction pathways. The NO-cGMPase-cGMP signaling pathway undertakes a wide range of functions affecting key biological processes, including neurotransmission, muscle relaxation, and inflammation, as well as host defense in vertebrate systems [34]. There are also data showing that such simple molecules are involved in regulating various biological processes in invertebrate systems [35]. The pathway of this signaling pathway is primarily the origin of endogenous biosynthesis of NO from a family of NO synthases (NOS), which converts L-arginine to NO. *cGMPase* is then activated by NO, and *cGMPase* catalyzes the conversion of GTP to cGMP, thus activating downstream components and effectors. The downstream cgmp-gated ion channels, PKG and PDE of phosphodiesterase protein kinase G, are activated and signaling cascades to cellular effectors [36]. Zhang et al. [37] found that the *cGMPase* inhibitor ODQ inhibits the biosynthesis of cGMP by inhibiting the *cGMPase* activity, thus accelerating larval attachment and metaporation. By inhibiting *cGMPase* activity, different researchers have found that this enzyme plays a major role in regulating metamorphosis development in the NO-cGMPase-cGMP pathway, and they have found that cGMP is involved in larval metamorphosis in several marine invertebrates [38,39].

*cGMPase*, when activated, converts GTP into cGMP. This may lead to further downstream reactions within cGMP-gated ion channels where phosphodiesterases (PDEs) or protein kinase G (PKG) may inhibit metamorphosis [37,40,41,42,43]. At the same time, studies have shown that *cGMPase* is regulated by NO, and NO [44] has a negative regulatory effect on apoptosis, which is an important process in metamorphosis and can lead to the loss of excess larval organs. In addition, activation of the mitogen-activated protein kinase/extracellular signal-regulated kinase (MAPK/ERK) pathway and subsequent regulation of metatoriation-related genes in species with NO-inducible effects [45,46].

In addition, it has been proved in mammals that *cGMPase* can participate in muscle development, especially smooth muscle development [47], while the muscles of invertebrates are mainly composed of smooth muscle with thick and fine muscle filaments. Therefore, before this study, we suspected that the muscle development process of *S. constricta* might be regulated by *cGMPase* during the metamorphosis.

The experiments used to determine the effective concentrations of dsRNA in the TPH-DVE, DVE-UMBO, and UMBO-CRE in three periods were soaked interference experiment. Additionally, the results showed that compared with the blank group and the NC group, interference can significantly inhibit in three periods of abnormal rate, and higher concentrations of it will be easier to kill. In the DVE stage when compared with the blank group, mortality increased significantly, but when compared with the NC group, no significant differences were observed, indicating that the EGFP synthesis process-related reagents may have a certain influence on the larval metamorphosis process in DVE and CRE, compared with the blank group and NC group. After 50 days, the inhibition rates of TPH-DVE and UMBO-CRE were 53% and 66%, respectively, and suggested that *cGMPase* had important roles during *S. constricta* growth and metamorphosis.

In dsRNA interference groups, mortality was significantly increased, suggesting larval adaptation was stronger at later metamorphosis stages. To explore if *cGMPase* was involved in regulating *S. constricta* growth, we performed in vivo dsRNA injection experiments into living razor clams. First, *cGMPase* expression levels at different time points were measured to determine changes in related growth traits, including shell length and body weight during long-term growth. Our results showed that delayed growth features were caused by long-term interference with *cGMPase*.

## 5. Conclusions

The goal of this study was to explore the changes of *cGMPase* gene during the growth and development of sinonovacula constricta by sequence analysis and dsRNA interference data. *cGMPase* plays an important role in the growth and development of *S. constricta* as a signal transduction molecule. *cGMPase* expression was identified in all tissues, with higher expression levels in the gills and liver. After *cGMPase* inhibition with dsRNA interference, larval metamorphosis and survival rates were considerably affected and suggested *cGMPase* inhibited shell length and body weight during juvenile growth periods. Thus, we verified the inhibitory effects of a growth-related signal transduction factor on razor clam growth and development. The results provided certain data reference for the study of the growth and development mechanism of shellfish. At the same time, in the future breeding of the family, we can detect the gene expression and other data to select the good varieties for the next breeding work. Based on this study, we will further study the specific regulatory mechanism of the NO-cGMPase-cGMP pathway on the growth and development of *S. constricta*. In summary, *cGMPase* provides an important genetic reference point for the breeding and seedling breeding of *S. constricta*.

## Figures and Tables

**Figure 1 genes-14-00459-f001:**
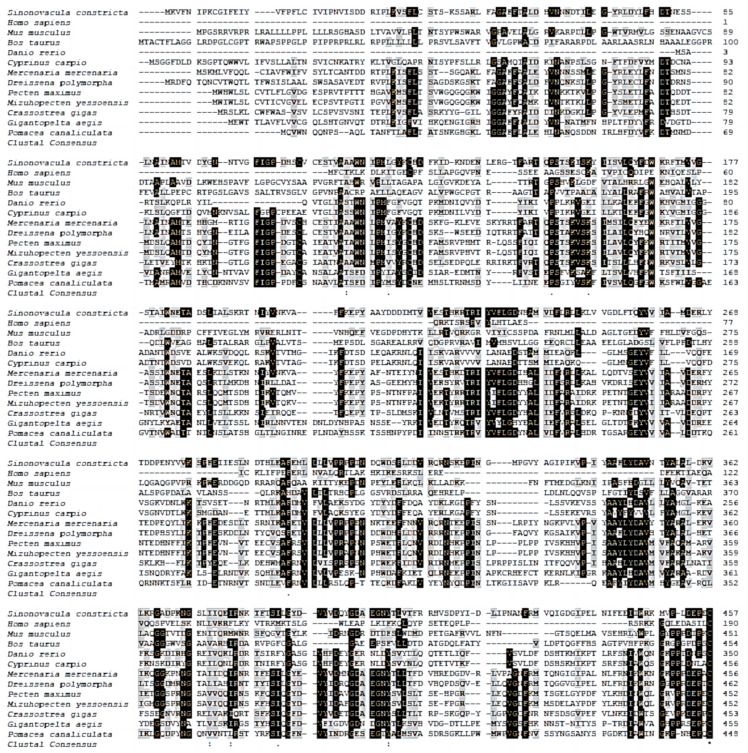
Multiple *cGMPase* amino acid sequence alignments from *S. constricta* and other species. Identical amino acid residues are highlighted in black and similar amino acids in gray. Accession numbers and species: *Homo sapiens*, CAA47145.1; *Mus musculus*, CAC41350.1; *Bos taurus*, AAA50790.1; *Danio rerio*, XP_009305068.1; *Cyprinus carpio*, XP_009305068.1; *M. mercenaria*, XP_045180432.1; *Dreissena polymorpha*, KAH3827983.1; *P. maximus*, XP_033751463.1; *Mizuhopecten yessoensis*, XP_021361670.1; *Crassostrea gigas*, XP_034317886.1; *Gigantopelta aegis*, XP_041362492.1; and *Pomacea analiculate*, XP_025089642.1. * *p* < 0.05, ** *p* < 0.01.

**Figure 2 genes-14-00459-f002:**
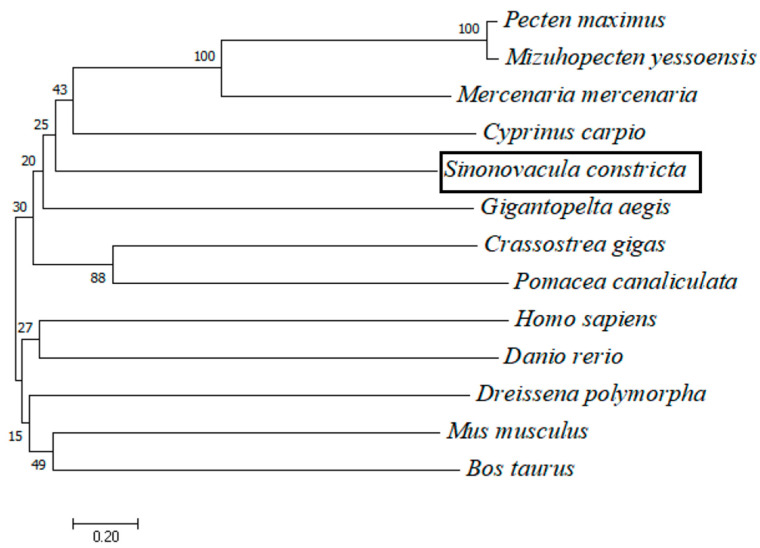
NJ phylogenetic tree of amino acid sequences from *cGMPase* in different species. Accession numbers and species: *P. maximus*, XP_033751463.1; *M. yessoensis*, XP_021361670.1; *M. mercenaria*, XP_045180432.1; *C. carpio*, XP_009305068.1; *G. aegis*, XP_041362492.1; *C. gigas*, XP_034317886.1; *P. analiculate*, XP_025089642.1; *H. sapiens*, CAA47145.1; *D. rerio*, XP_009305068.1; *D. polymorpha*, KAH3827983.1; *M. musculus*, CAC41350.1; and *B. taurus*, AAA50790.1.

**Figure 3 genes-14-00459-f003:**
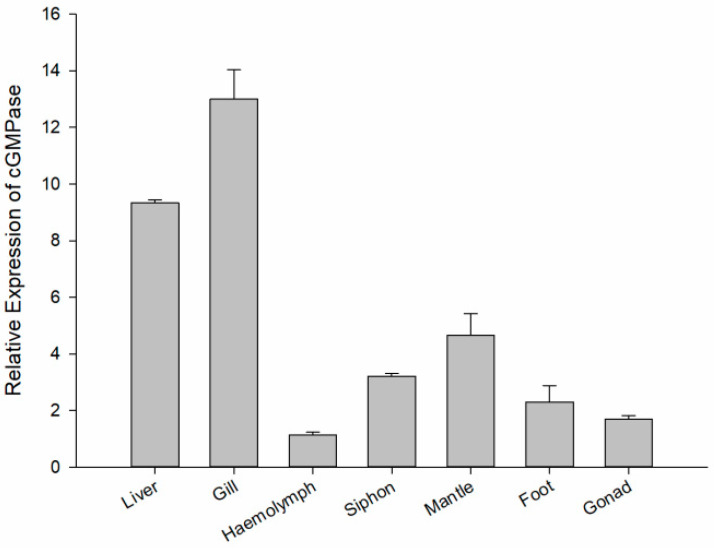
Expression analysis showing *cGMPase* expression profiles in different tissue, including liver, gill, haemolymph, siphon, mantle, foot, and gonad.

**Figure 4 genes-14-00459-f004:**
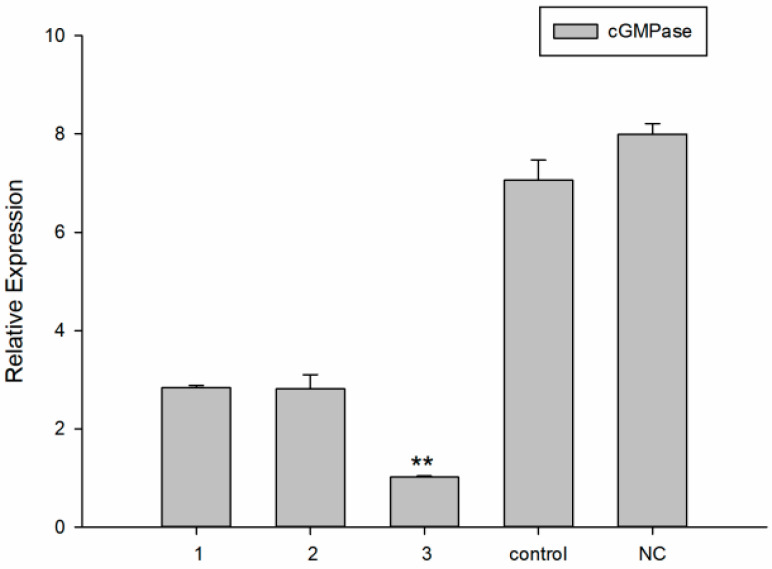
Screening for effective dsRNAs of *cGMPase*. Control was the blank group, and NC was the negative control group. Groups 1, 2, and 3 were the three chains. Double asterisks mean a highly significant difference (** *p* < 0.01).

**Figure 5 genes-14-00459-f005:**
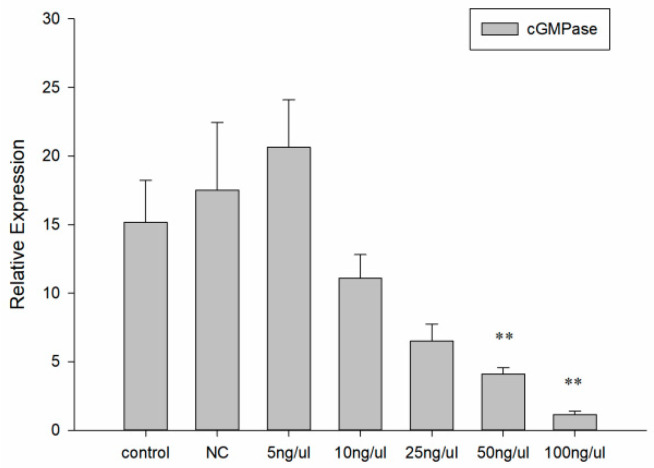
Screening of effective dsRNA action concentration of *cGMPase*. Control group was blank group, NC group was negative control group. Double asterisks mean a highly significant difference (** *p* < 0.01).

**Figure 6 genes-14-00459-f006:**
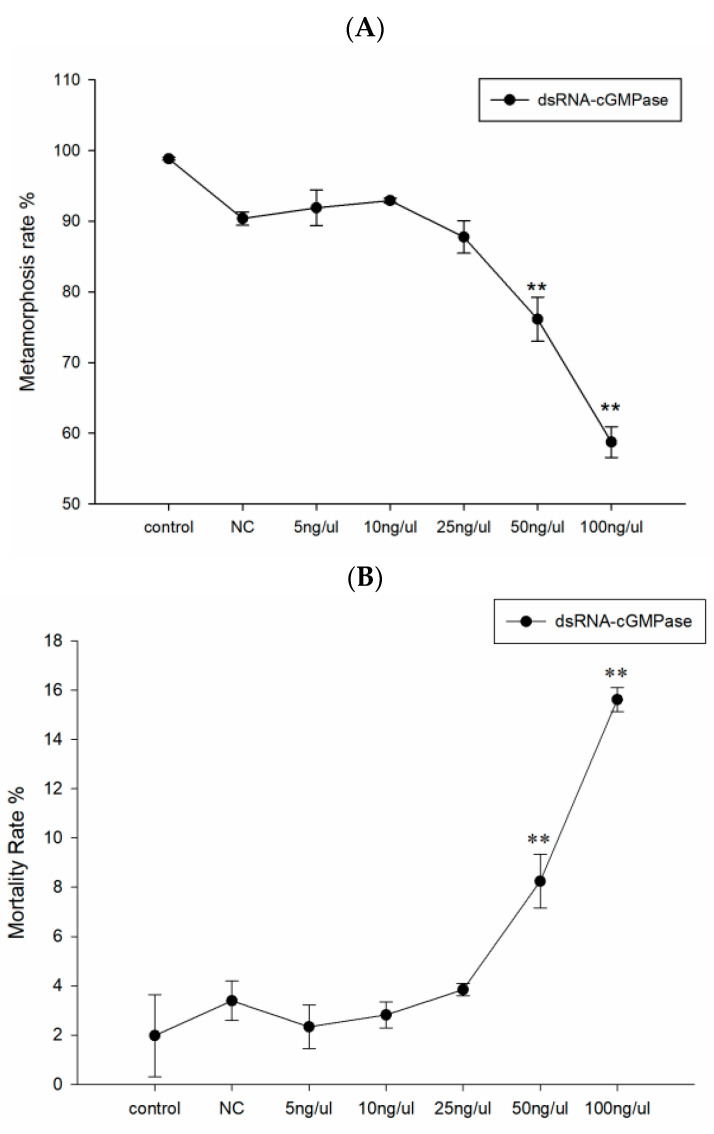
During the DVE-UMBO, different dsRNA concentrations interfered with metamorphosis (**A**) and mortality rates (**B**) of *cGMPase* in larvae. Double asterisks mean a highly significant difference (** *p* < 0.01).

**Figure 7 genes-14-00459-f007:**
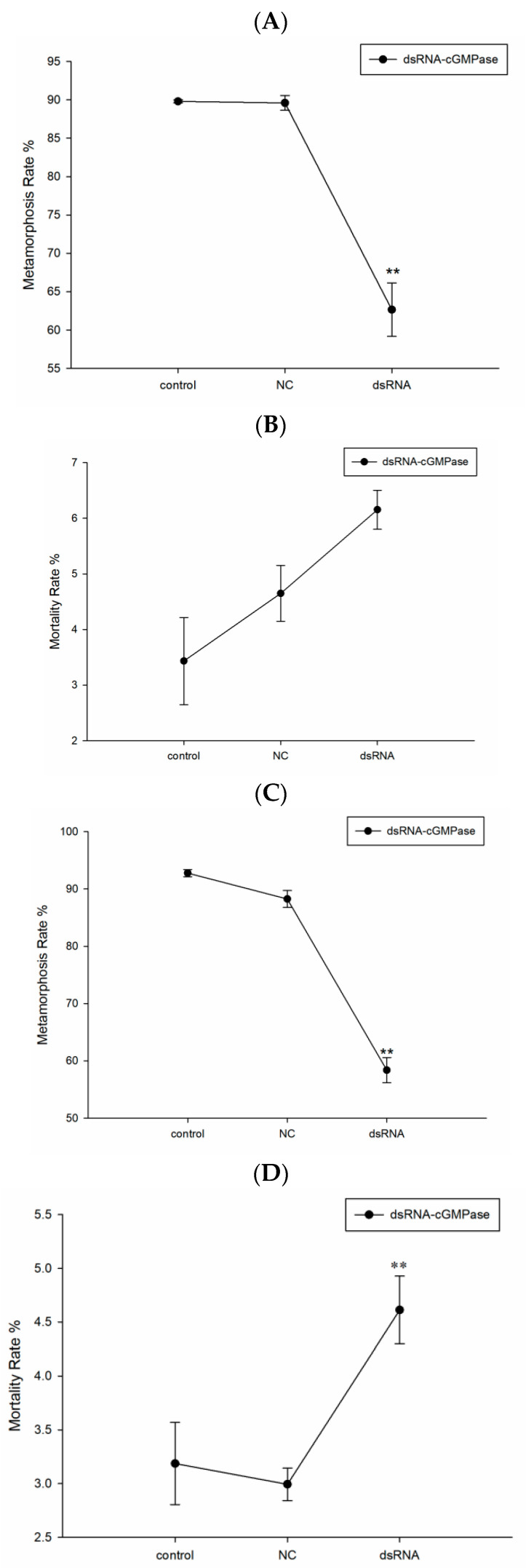
Metamorphosis and mortality rates in larvae during dsRNA interference against *cGMPase* (**A**–**D**) at TPH-DVE and UMBO-CRE stages: (**A**,**B**) metamorphosis and mortality rates in larvae at the TPH-DVE stage, respectively; (**C**,**D**) metamorphosis and mortality rates in larvae at the UMBO-CRE stage, respectively. Double asterisks mean a highly significant difference (** *p* < 0.01).

**Figure 8 genes-14-00459-f008:**
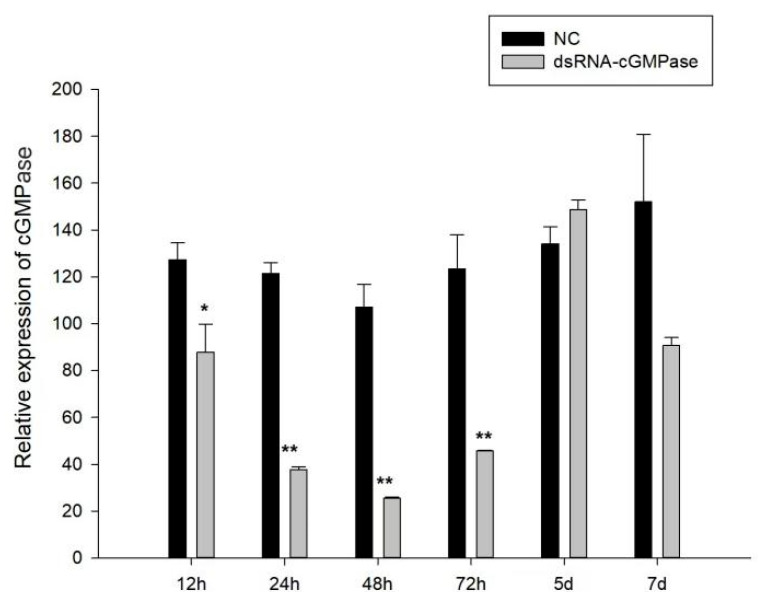
Screening dsRNA interference toward *cGMPase* time points during the *S. constricta* growth period. The horizontal axis represents different time points after injection. A single asterisk means there is a signifificant difference (* *p* < 0.05), and double asterisks mean a highly significant difference (** *p* < 0.01).

**Figure 9 genes-14-00459-f009:**
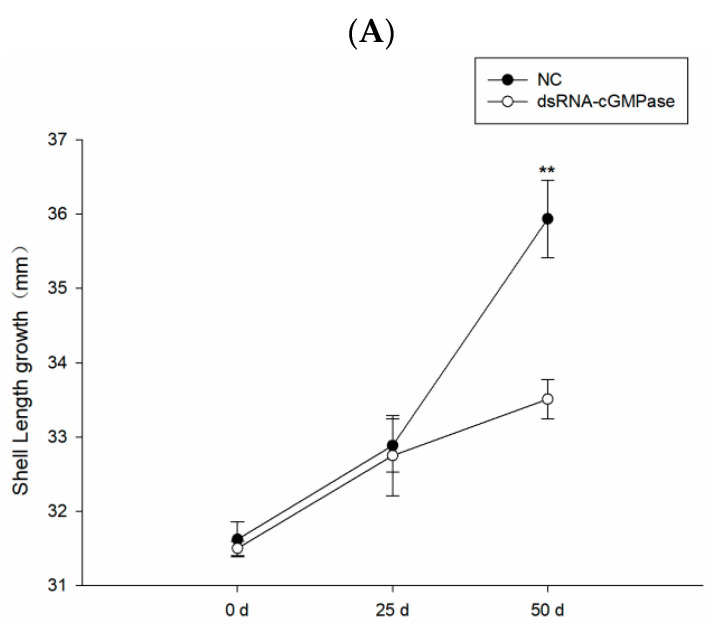
*S. constricta* growth after the long-term injection of dsRNA-*cGMPase*. Shell length (**A**) and weight gain (**B**) at 0, 25, and 50 days after injection. A single asterisk means there is a signifificant difference (* *p* < 0.05), and double asterisks mean a highly significant difference (** *p* < 0.01).

**Table 1 genes-14-00459-t001:** Primers used for sequence identification.

Primers Name	Sequence (5′–3′)	Purpose
M13-F	AGGGTTTTCCCAGTCACG	Sequence verification of *cGMPase*
M13-R	AGCGGATAACAATTTCACAC	
*cGMPase* F1	TGCCTTCCTGTATGATGCT	
*cGMPase* R1	TTTCAGTAACTCAGGTGCC	
*cGMPase* F2	CTGTTAGGAAAAAGAGTCG	
*cGMPase* R2	GCTACAGATCCTGCTTATT	
*cGMPase* F3	ATGATGAGTAGATGACCCC	
*cGMPase* R3	CATCATACAGGAAGGCAGC	
18S-F	TCGGTTCTATTGCGTTGGTTTT	qRT-PCR of control
18S-R	CAGTTGGCATCGTTTATGGTCA	

**Table 2 genes-14-00459-t002:** Primers used in the experiment process.

Name of Primer	Specific Sequence (5′–3′)
*cGMPase* F1	GCGACGTGTATTCTTTCGCC
*cGMPase* R1	CAACGCAGTGAAGCCAACAA
*cGMPase* F2	ACTACGACCGATTGCTGTGG
*cGMPase* R2	TGTCCTCTCCGCAACAAGAC
*cGMPase* F3	AACCGGATTTCCGACCTAGC
*cGMPase* R3	TTCTCCACGATGGCGTCAAA
*cGMPase* T7 F1	TAATACGACTCACTATAGGGGCGACGTGTATTCTTTCGCC
*cGMPase* T7 R1	TAATACGACTCACTATAGGGCAACGCAGTGAAGCCAACAA
*cGMPase* T7 F2	TAATACGACTCACTATAGGGACTACGACCGATTGCTGTGG
*cGMPase* T7 R2	TAATACGACTCACTATAGGGTGTCCTCTCCGCAACAAGAC
*cGMPase* T7 F3	TAATACGACTCACTATAGGGAACCGGATTTCCGACCTAGC
*cGMPase* T7 R3	TAATACGACTCACTATAGGGTTCTCCACGATGGCGTCAAA
EGFP-F	CAGTGCTTCAGCCGCTACC
EGFP-R	AGTTCACCTTGATGCCGTTCTT

## Data Availability

All experimental data used for supporting the conclusions in this article are reasonably available through the authors.

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
