# Peer review of "The Inhibitory Effects of RNA-Interference-Mediated Guanylate Cyclase Knockdown on Larval Metamorphosis and Early Progeny Growth of Razor Clam"

_genes, 2023, doi:10.3390/genes14020459_

Round 1

Reviewer 2 Report

This paper is very interesting, and it can be accepted after revision. However, I have several concerned questions:

1. Title should revise it to “The inhibitory effects of RNA interference-mediated guanylate cyclase knockdown on larval metamorphosis and early progeny growth of razor clam”.

2. Line 164-172, six well plates, How much of each hole size, eg, 5ml, or 10ml, and How much of larvae rearing density?

3. How long time to change one time for dsRNAs solution, and how to keep  concentrations of effective dsRNAs?

4. How to examined the effects of gene knockdown? Especial to those of small larvae.

5. Result is that the inhibitory effects of RNA interference-mediated guanylate cyclase knockdown in razor clam is significant, how to apply to breeding?
